# Gradient-independent Wnt signaling instructs asymmetric neurite pruning in *C. elegans*

Menghao Lu[1], Kota Mizumoto[1,2]*

[1]Department of Zoology, University of British Columbia, Vancouver, Canada; [2]Life Sciences Institute, University of British Columbia, Vancouver, Canada

**Abstract** During development, the nervous system undergoes a refinement process by which neurons initially extend an excess number of neurites, the majority of which will be eliminated by the mechanism called neurite pruning. Some neurites undergo stereotyped and developmentally regulated pruning. However, the signaling cues that instruct stereotyped neurite pruning are yet to be fully elucidated. Here we show that Wnt morphogen instructs stereotyped neurite pruning for proper neurite projection patterning of the cholinergic motor neuron called PDB in *C. elegans*. In *lin-44/wnt* and *lin-17/frizzled* mutant animals, the PDB neurites often failed to prune and grew towards the *lin-44*-expressing cells. Surprisingly, membrane-tethered *lin-44* is sufficient to induce proper neurite pruning in PDB, suggesting that neurite pruning does not require a Wnt gradient. LIN-17 and DSH-1/Dishevelled proteins were recruited to the pruning neurites in *lin-44*-dependent manners. Our results revealed the novel gradient-independent role of Wnt signaling in instructing neurite pruning.

## Introduction

Development of the nervous system is a highly dynamic process which involves neurogenesis, cell migration, neuronal polarization and outgrowth of neuronal processes. It is also well known that neurons initially extend an excess number of neurites most of which will later be eliminated by the mechanism called neurite pruning (*Riccomagno and Kolodkin, 2015*; *Schuldiner and Yaron, 2015*). Failure in neurite pruning results in excess arborization and neuronal connectivity, which could underlie various neurological conditions including macrocephaly (*Jan and Jan, 2010*). In vivo imaging techniques have allowed for the elucidation of genetic and molecular mechanisms that underlie neuronal pruning.

During neuronal refinement, neurites often compete with each other for their target cells in an activity-dependent manner, and neurites that are outcompeted are eliminated by pruning. For example, each muscle fiber in the mouse trapezius muscle is initially innervated by multiple motor neurons in early postnatal stage, while axons with weaker synaptic strength are pruned by the mechanism called axosome shedding (*Bishop et al., 2004*; *Colman et al., 1997*). The neurites also compete for trophic factors. Axons from rodent sympathetic neurons receive nerve growth factor (NGF) secreted from the target cells for their survival by inducing the brain-derived neurotrophic factor (BDNF), which in turn induces pruning of the outcompeted axons (*Deppmann et al., 2008*; *Singh et al., 2008*).

In addition to activity-dependent pruning, many neurites undergo stereotyped pruning during development (*Schuldiner and Yaron, 2015*). Several signaling molecules have been shown to play critical roles in this process. In the mouse hippocampus, a chemorepellent Semaphorin 3F (Sema3F) is expressed in the distal infrapyramidal region and locally induces infrapyramidal tract (IFT) axon pruning via the PlexinA3-Neuropilin2 receptor complex (*Bagri et al., 2003*; *Sahay et al., 2003*).

*For correspondence:
mizumoto@zoology.ubc.ca

Competing interests: The authors declare that no competing interests exist.

Neuropilin2 binds and activates β2-Chimaerin, a RacGAP, to inactivate Rac1 GTPase to induce IFT axon pruning (*Riccomagno et al., 2012*). Interestingly, loss of β2-Chimaerin specifically affects the pruning of the IFT axons but does not mimic the other phenotypes of the Sema3F or PlexinA2 such as axon guidance. Therefore, Sema-Plexin may utilize a specific downstream signaling cascade to control neurite pruning that differs from its roles in axon guidance and synapse formation.

Neurite pruning also plays a crucial role in sexually dimorphic neurocircuit formation. The expression of BDNF from the mammary mesenchyme is required for the innervation of the mammary gland sensory axons. In males but not in females, a truncated TrkB receptor isoform is expressed in the mammary mesenchyme which deprives BDNF availability for sensory axons thereby inducing male-specific axon pruning (*Liu et al., 2012*). Recent work revealed that Sema-Plexin signaling acts as a pruning cue for these sensory axons, as Sema6A and Sema3D expression in the mammary gland act to induce sensory axon pruning through PlexinA4 (*Sar Shalom et al., 2019*).

Wnt is a secreted morphogen whose gradient distribution plays critical roles in various steps of animal development (*Bartscherer and Boutros, 2008*; *Yang, 2012*). Wnt signaling is also crucial in the nervous system development including neurogenesis, neuronal polarization, neurite outgrowth, axon guidance and synapse formation (*He et al., 2018*; *Inestrosa and Varela-Nallar, 2015*; *Park and Shen, 2012*). Nevertheless, its implication in neurite pruning is still limited. In *Caenorhabditis elegans*, it has been shown that the transcription factor MBR-1/Mblk-1 is required for the pruning of the excess neurites of the AIM interneurons (*Kage et al., 2005*), whose function is antagonized by the trophic role of Wnt signaling (*Hayashi et al., 2009*). It has also been shown that Wnt signaling functions as a 'pro-retraction' cue in the Drosophila contralaterally-projecting serotonin-immunoreactive deuterocerebral interneurons (CSDns) (*Singh et al., 2010*). It is yet to be determined whether Wnt signaling plays instructive or permissive roles in neurite pruning.

In the present study, we describe a gradient-independent Wnt signaling in instructing stereotyped developmental neurite pruning of the PDB cholinergic motor neuron in *C. elegans*. From genetic analyses, we found that expression of LIN-44/Wnt in the tail hypodermal cells instructs pruning of the PDB neurites growing towards the tail hypodermal cells. Surprisingly, membrane-tethered LIN-44/Wnt is sufficient to induce PDB neurite pruning, suggesting that the gradient distribution of LIN-44 is not essential for neurite pruning. During this process, LIN-44/Wnt directs asymmetric localization of its receptor LIN-17/Frizzled and the intracellular signaling component DSH-1/Dishevelled to the posterior pruning neurites. Taken together, we discovered the novel gradient-independent Wnt signaling in instructing developmental neurite pruning.

## Results

### Asymmetric neurite pruning during PDB development

PDB is a post-embryonic cholinergic motor neuron derived from the P12 neuroectoblast cell (*Sulston and Horvitz, 1977*), and its cell body resides in the preanal ganglion. In hermaphrodites, PDB sends a single neurite posteriorly toward the tail tip region, where it makes a sharp 'V-shape' turn to join the dorsal nerve cord to form *en passant* synapses onto dorsal body wall muscles (*Figure 1A*). While the function of PDB has not been extensively studied, a recent study combining mathematical modeling and a cell ablation experiment revealed the role of PDB in motor coordination (*Yan et al., 2017*). To investigate the PDB neurite development, we first generated a strain in which the PDB neurite and presynapses are labeled with a transgene (*mizIs9*) that expresses a membrane-tethered GFPnovo2, a brighter GFP variant (*Hendi and Mizumoto, 2018*), and a synaptic vesicle gene, *rab-3*, fused with mCherry under the *kal-1* promoter (*Bülow et al., 2002*) (see Materials and methods) (*Figure 1A*).

The PDB neuron extends its single neurite during late first larval (L1) and second larval (L2) stages. To understand how PDB neurite achieves V-shape turning, we first examined the structure of the developing PDB neurite using a population of semi-synchronized animals (*Figure 1B*). Interestingly, we observed that the PDB neuron has a unique transient structure at L2 stage: the ventral neurite sends a tiny commissure to the dorsal side of the worm where it bifurcates to extend one neurite posteriorly and one anteriorly (*Figure 1C*: upper panels). The PDB neuron therefore often has 'H-shape' structure with two posterior neurites (one dorsal neurite and one ventral neurite) at the early L2 stage (*Figure 1C and E*). In sharp contrast, the PDB neuron of most L3/L4 animals contained no

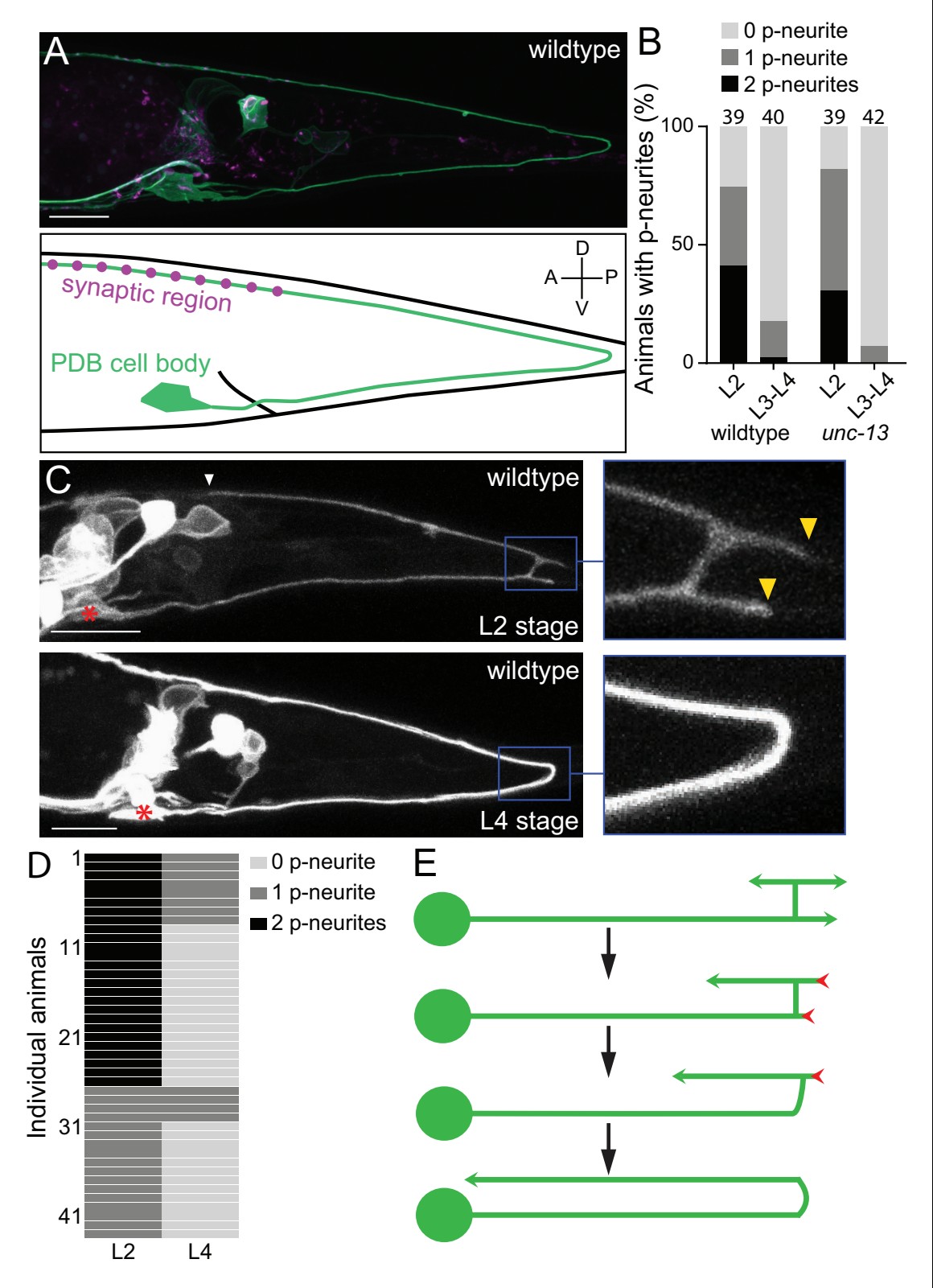

**Figure 1.** PDB neurites undergo stereotyped asymmetric pruning during development. (**A**) Structure of PDB labeled by the *mizIs9* transgene. The PDB process and presynaptic sites are labeled with GFPnovo2::CAAX (green) and mCherry::RAB-3 presynaptic vesicle marker (magenta), respectively. A schematic is shown in the bottom panel. (**B**) Quantification of the number of posterior neurites (p-neurites) in semi-synchronized populations of wildtype and *unc-13* mutants at L2 and L4 stages respectively. Sample number is indicated above each bar (**C**) Representative images of the posterior neurite

*Figure 1 continued on next page*

*Figure 1 continued*

pruning event in a wildtype animal at L2 and L4 stages. The regions of posterior neurites are magnified in the right panels. Asterisks represent the PDB cell body. White and yellow arrowheads denote the end of anterior and posterior neurites, respectively. (D) Quantification of the posterior neurite number of 43 wildtype animals at L2 and L4 stages. (E) A schematic of asymmetric neurite pruning during PDB development. Green and red arrowheads represent growing and pruning neurites. Scale bars: 10 μm.

The online version of this article includes the following source data and figure supplement(s) for figure 1:

**Source data 1.** Quantification of the number of posterior neurites.

**Figure supplement 1.** Time-lapse imaging of PDB neurite pruning.

posterior neurite (*Figure 1A and B*). This observation suggests that the posterior neurites observed at the L2 stage are the transient structures most of which are pruned by the L4 stage. To confirm that the posterior neurites that are present at L2 stage are indeed pruned during development, we examined the PDB structure of single animal at two developmental time points, second larval (L2) and last larval (L4) stages. We selected 43 L2 animals with at least one posterior neurite in PDB and re-examined its structure at L3/L4 stage (24 hr after the first examination) (*Figure 1C and D*). Among 26 animals that had two neurites at the L2 stage, 18 animals had no posterior neurite, and eight animals had one neurite at L3/L4 stage. Among 17 animals with one posterior neurite at the L2 stage, 13 animals had no posterior neurite and four animals had one neurite at L3/L4 stage. In total, 82.6% (57/69) of the neurites we observed at the L2 stage were pruned within 24 hr.

Neurons could prune their neurites either by retracting or severing them (*Schuldiner and Yaron, 2015*). To distinguish which mechanism the PDB neuron utilizes to prune its posterior neurites, we conducted time-lapse imaging during posterior neurite pruning and found that the posterior neurite became shorter over the development while the dorsal anterior neurite kept growing (*Figure 1—figure supplement 1*). This indicates that the posterior neurites are pruned by retraction rather than the severing followed by degradation. To determine if PDB pruning is dependent on neuronal activity, we examined mutants of *unc-13*, which is required for synaptic vesicle fusion and neurotransmitter release (*Richmond et al., 1999*). At L2 stage of *unc-13* animals, we observed PDB posterior neurites at a similar frequency as wildtype, while most of the *unc-13* animals at late larval stage (L3-L4) did not have PDB posterior neurites as observed in wildtype (*Figure 1B*). Consistently, we did not observe significant structural defects in PDB of *unc-13* mutants at L4 stage (*Figure 2D*), suggesting that the PDB neurite pruning is not dependent on the neuronal activity. Taken together, our observation indicates that the V-shape turn of the PDB neurite is achieved by the asymmetric pruning of the posterior but not the anterior neurites at the turning point during PDB development (*Figure 1E*).

## LIN-44/Wnt is required for the asymmetric neurite pruning in PDB

We next sought to identify the molecular cue that instructs pruning of the posterior neurites in PDB. Recent work has shown that the guidance and outgrowth of the PDB neurite are disrupted in the mutants of Wnt signaling and syndecan proteoglycans (*Saied-Santiago et al., 2017*). Wnt is a family of highly conserved secreted glycoproteins that play pivotal roles in animal development (*Nusse, 2005*). Wnt acts through receptors including Frizzled, LRP5/6 and receptor *tyrosine* kinase (Ryk). Upon Wnt binding, these receptors execute distinct downstream cascades via the multidomain protein, Dishevelled (*Sawa and Korswagen, 2013*; *van Amerongen and Nusse, 2009*).

Previous works have shown that *lin-44/wnt* is expressed in the hypodermal cells at the tail tip region (*Figure 2A*), where it locally inhibits neurite outgrowth and synapse formation, as well as controls neuronal polarity and asymmetric cell division (*Herman et al., 1995*; *Hilliard and Bargmann, 2006*; *Klassen and Shen, 2007*; *Maro et al., 2009*; *Zheng et al., 2015*). We therefore hypothesized that *lin-44/wnt* functions as a pruning cue for the PDB posterior neurites. To test this hypothesis, we first examined the relative position of the PDB neurites and the *lin-44*-expressing cells labeled with the blue fluorescent protein (BFP) expressed under the *lin-44* promoter during PDB development. Strikingly, at the L2 stage when PDB posterior neurites undergo pruning, we found that the pruning posterior neurite was located in close proximity to the most posterior hypodermal cell, hyp10 (*Figure 2B–B' and C*). Interestingly, we noticed that hyp10 often showed the brightest BFP expression among the *lin-44*-expressing cells at the L2 stage. The position of the PDB posterior neurites

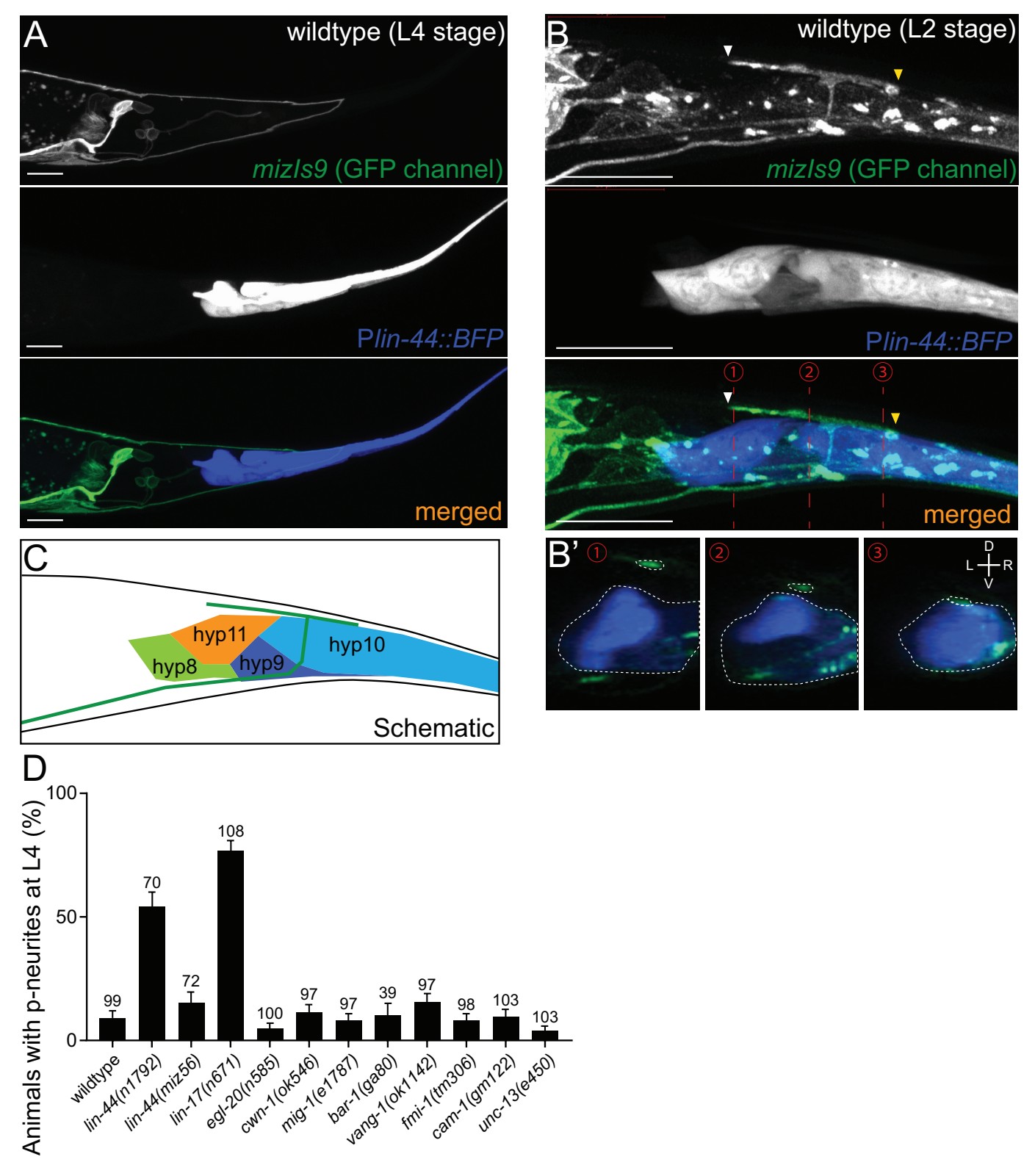

**Figure 2.** *lin-44/wnt* expressed adjacent to PDB posterior neurites is required for PDB development. (**A and B**) Representative images of PDB neurite labeled with *mizIs9* (top panels), *lin-44*-expressing cells labeled with P*lin-44::BFP* (middle panels) and merged images (bottom panels) at the L4 stage (**A**) and during posterior neurite pruning at the L2 stage (**B**). White and yellow arrowheads denote anterior and posterior neurites respectively. (**B'**) The transverse section of three positions of PDB neurites (indicated by red dotted lines in **B**) are reconstituted from the z-stack images shown in **B**. Dotted

*Figure 2 continued on next page*

*Figure 2 continued*

circles highlight PDB neurites and *lin-44*-expressing cells. (**C**) A schematic of (**B**). Green line represents PDB neurites. (**D**) Quantification of the animals with posterior PDB neurites at L4 stage. Animals with defects in PDB cell fate specification and neurite guidance were excluded from this quantification (see *Figure 2—figure supplement 1*). Error bars represent standard error of proportion (SEP). Sample numbers are shown above each bar. Scale bars: 10 μm.

The online version of this article includes the following source data and figure supplement(s) for figure 2:

**Source data 1.** Quantification of the animals with posterior PDB neurites.

**Figure supplement 1.** Cell fate and guidance defects of PDB in the mutants of Wnt signaling components.

**Figure supplement 1—source data 1.** Quantification of the animals with defects in PDB cell fate specification and neurite guidance.

and hyp10 which expressed a high level of *lin-44* is well in line with our prediction that *lin-44/wnt* is the pruning cue. We then examined whether the pruning of the posterior neurites is affected in *lin-44* null mutants. At L4 stage, 24% (24/100) and 6% (6/100) of the *lin-44(n1792)* null animals showed neurite guidance and cell fate determination defects (no PDB), respectively (*Figure 2—figure supplement 1*). We excluded them from our quantification of the neurite pruning in the following experiments because these phenotypes prevented us from examining the pruning defects. Among the *lin-44* mutant animals with no obvious cell fate or guidance defects, 54.3% (38/70) of them showed ectopic posterior neurites in PDB at the L4 stage, suggesting the defective neurite pruning during PDB development (*Figure 2D*). To test if the pruning of the posterior neurites is compromised in *lin-44* mutants, we compared the PDB structure of the same animal at L2 and L4 stages (*Figure 3A and D*). As expected, we indeed observed that only 30% (15/50) of the posterior neurites observed at the L2 stage animals are pruned by L4 stage in the *lin-44* mutants, which is significantly lower than those observed in wildtype (82.6%) (*Figure 3C and D*). This result strongly suggests that *lin-44/wnt* is required for the pruning of the posterior neurites during PDB development.

## *lin-17/fz* acts cell-autonomously in PDB to induce neurite pruning

LIN-44/Wnt utilizes LIN-17/Frizzled(Fz) as a primary receptor in many contexts (*Hilliard and Bargmann, 2006*; *Klassen and Shen, 2007*; *Sawa et al., 1996*). We then tested if *lin-17/fz* also plays a role in neurite pruning in PDB. While *lin-17(n671)* null mutants showed significantly more severe cell fate or guidance defects than *lin-44/wnt* mutants (*Figure 2—figure supplement 1*), we observed similar posterior neurite pruning defects as *lin-44* mutants: only 17.0% of the posterior neurites (9/53) were pruned 24 hr after L2 stage (*Figure 3B–3D*). The pruning defect in the *lin-44; lin-17* double mutants (14.3%: 6/42 of the posterior neurites were pruned) was not significantly different from *lin-17* single mutants, suggesting that *lin-44* and *lin-17* act in the same genetic pathway (*Figure 3C and D*). We also conducted tissue-specific rescue experiments to test if *lin-17* functions cell-autonomously in PDB. Expression of *lin-17* cDNA either from the pan-neuronal promoter (P*rgef-1*) or the PDB promoter (P*kal-1*) significantly rescued the pruning defects of *lin-17* mutants, suggesting that *lin-17* acts cell-autonomously in PDB (*Figure 3—figure supplement 1*). These results indicate that LIN-17/Fz is a receptor that functions in PDB to receive LIN-44/Wnt signal to induce neurite pruning.

## LIN-44/Wnt but not EGL-20/Wnt instructs PDB neurite pruning

Since *lin-44/wnt* is expressed posteriorly to the PDB neurites, it is possible that LIN-44 acts as an instructive cue for PDB neurite pruning. Alternatively, it might also act as a permissive cue for PDB neurite pruning. In order to distinguish these two possibilities, we conducted the rescue experiment by expressing LIN-44 ectopically using the promoter of another *wnt* gene, *egl-20*. *egl-20/wnt* is expressed in the cells around preanal ganglions (*Whangbo and Kenyon, 1999*). If LIN-44/Wnt functions as an instructive cue, its expression from anterior cells using the *egl-20* promoter would not rescue the PDB pruning defect in *lin-44* null mutants. If it acts as a permissive cue, we expect to see the rescue of the PDB neurite pruning defects no matter where *lin-44* is expressed. We found that P*egl-20::lin-44* transgenes did not rescue pruning defects of *lin-44* null mutants (*Figure 4A and B*). Similarly, ectopic anterior expression of *lin-44* from the dorsal body wall muscles using the truncated *unc-129* promoter (P*unc-129dm*) (*Colavita et al., 1998*) was not sufficient to rescue the PDB pruning defect in *lin-44* null mutants (*Figure 4A and B*). On the other hand, the transgenes expressing *lin-44* from the *lin-44* promoter (Ex[P*lin-44::lin-44*]) completely rescued the pruning defects to the wildtype

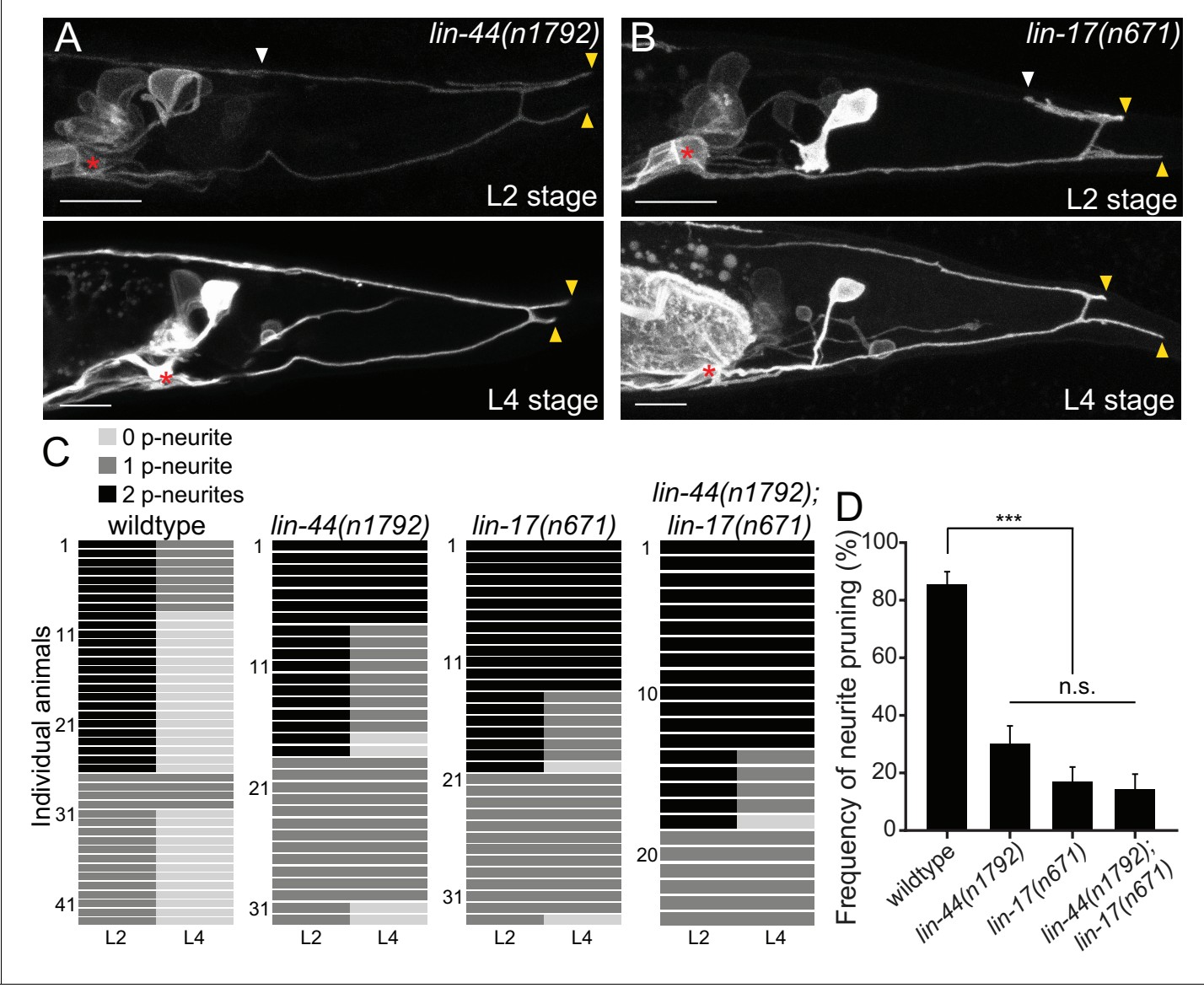

**Figure 3.** *lin-44/wnt* and *lin-17/fz* are required for the posterior neurite pruning. (**A and B**) Representative images of single animals of *lin-44(n1792)* (**A**) and *lin-17(n671)* (**B**) mutants at L2 (top panels) and L4 (bottom panels) stages. Asterisks represent PDB cell body. White and yellow arrowheads denote anterior and posterior neurites respectively. (**C**) Quantification of the posterior neurite number of individual animals at L2 and L4 stages in each genetic background. Note that quantification of wildtype is from *Figure 1*. (**D**) Quantification of the posterior neurite pruning frequency. ***p<0.001; n.s., not significant (Chi-square with Yates' correction). Error bars represent standard error of proportion (SEP). Scale bars: 10 μm.

The online version of this article includes the following source data and figure supplement(s) for figure 3:

**Source data 1.** Quantification of the posterior neurite pruning frequency.

**Figure supplement 1.** *lin-17/fz* acts cell-autonomously in PDB.

**Figure supplement 1—source data 1.** Quantification of posterior neurite pruning frequency.

level (*Figure 4A and B*). This result suggests that *lin-44/wnt* is an instructive cue for the posterior neurite pruning in PDB. *egl-20/wnt* often acts cooperatively and redundantly with *lin-44/wnt* in various developmental events (*Mizumoto and Shen, 2013*; *Yamamoto et al., 2011*). We therefore tested if *egl-20/wnt* can replace the function of *lin-44* in PDB neurite pruning. Interestingly, the expression of *egl-20* under the *lin-44* promoter did not rescue the PDB neurite pruning defects of the *lin-44* null mutant animals, suggesting the functional divergence between *lin-44* and *egl-20* in neurite pruning (*Figure 4A and B*). Consistently, PDB structure is largely unaffected in the mutants

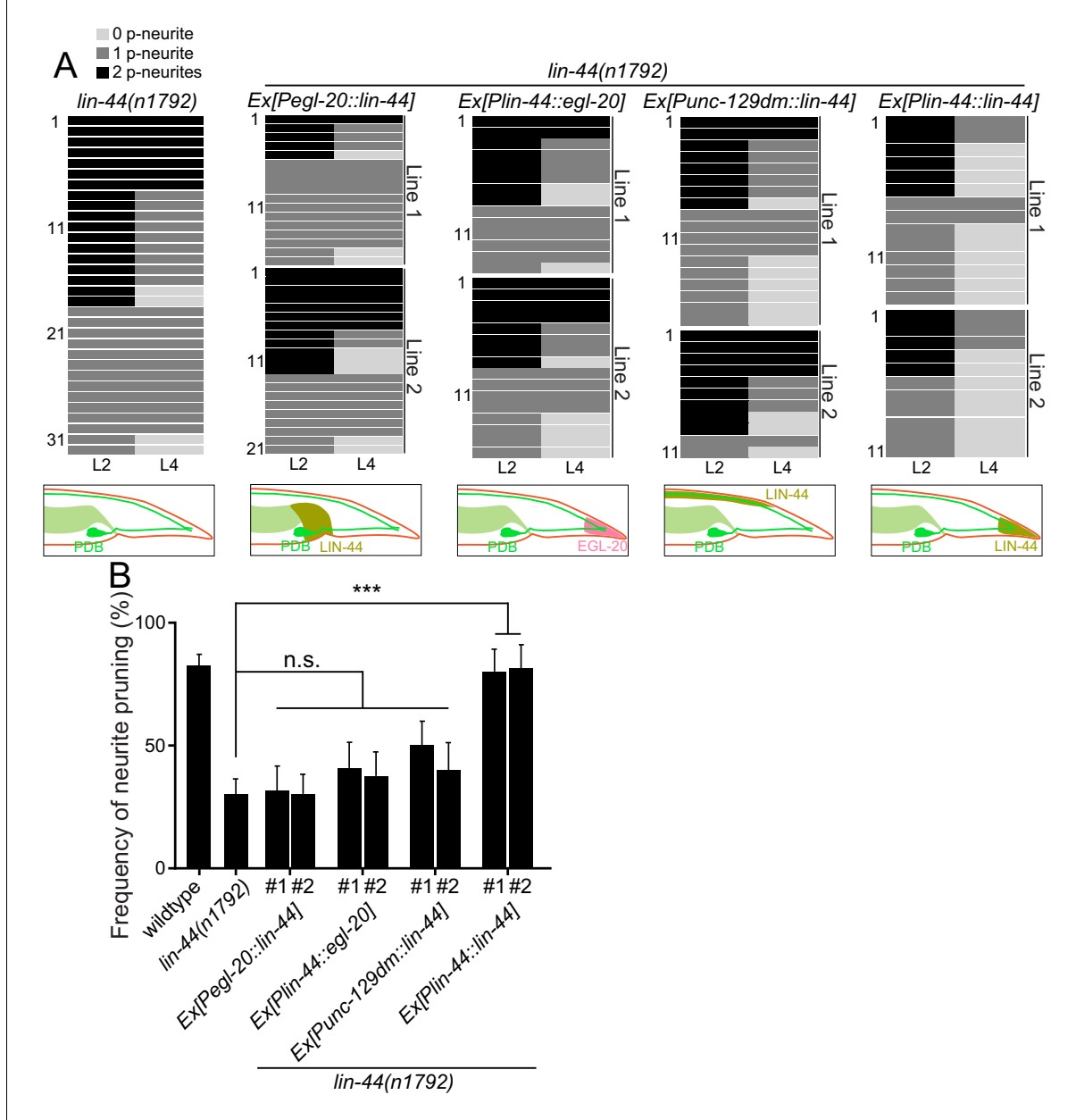

**Figure 4.** *lin-44* but not *egl-20* instructs neurite pruning in PDB. (**A**) Quantification of the posterior neurite numbers of individual animals at L2 and L4 stages in *lin-44* mutants and *lin-44* mutants with rescuing transgenes. Note that the quantification of *lin-44* mutants is from *Figure 3*. Bottom panels are schematics showing the expression domain of *lin-44* (green) and *egl-20* (magenta) in each genotype. Two independent transgenic lines are quantified (line #1 and line #2) for each rescuing construct. (**B**) Quantification of posterior neurite pruning frequency. ***p<0.001; n.s., not significant (Chi-square with Yates' correction). Error bars represent standard error of proportion (SEP).

The online version of this article includes the following source data for figure 4:

**Source data 1.** Quantification of posterior neurite pruning frequency.

of *egl-20(n585)* and its receptors *mig-1/fz* and *cam-1/Ror* (*Figure 2D* and data not shown) (*Eisenmann, 2005*; *Mizumoto and Shen, 2013*). Another Wnt gene, *cwn-1*, is also expressed in the posterior region of the worm (*Harterink et al., 2011*). The PDB morphology was indistinguishable from wildtype in the *cwn-1* mutants (*Figure 2D* and data not shown). While we do not completely exclude the involvement of other Wnts, our results indicate that LIN-44 is the major instructive cue for PDB neurite pruning.

## Membrane-tethered LIN-44 is sufficient to instruct PDB neurite pruning

Genetic experiments such as ectopic expression or overexpression of Wnts, and direct visualization of the Wnt protein have revealed the critical roles of gradient distribution of LIN-44 and EGL-20 in *C. elegans* (*Coudreuse et al., 2006*; *Klassen and Shen, 2007*; *Maro et al., 2009*; *Mizumoto and Shen, 2013*; *Pani and Goldstein, 2018*). While data from the above experiments are consistent with the idea that *lin-44/wnt* instructs asymmetric neurite pruning in PDB, several observations cannot be explained by the instructive role of LIN-44 as a gradient cue. First, we did not observe ectopic pruning of the anterior neurite when *lin-44/wnt* was expressed anteriorly from the *egl-20* promoter (data not shown), suggesting that anterior expression of *lin-44* is not sufficient to induce anterior neurite pruning in PDB. Second, during PDB development, both growing anterior and pruning posterior neurites are located close to the *lin-44*-expressing hyp10 cell (*Figure 2B*). If *lin-44/wnt* functions as a gradient pruning cue, how can the anterior neurite escape from the pruning signal? One possible explanation is that PDB neurites require high and locally concentrated LIN-44/Wnt to induce pruning rather than responding to the gradient distribution of LIN-44. In this case, the cells expressing *Pegl-20::lin-44* are too far away from the anterior neurite of PDB. Furthermore, a close examination of the spatial relationship between PDB neurites and the *lin-44*-expressing cells suggests that the GFP signal from the PDB posterior neurite tip overlaps with the BFP signal from hyp10 while there is a gap between the anterior neurite tip and hyp10 (*Figure 2B*). This raises one possibility that PDB neurites need to be in very close proximity, if not in contact, with the LIN-44-expressing cells to activate the pruning mechanism. To test this hypothesis, we took advantage of the membrane-tethered Wnt originally developed in *Drosophila* (*Zecca et al., 1996*). In *Drosophila*, the fusion construct of the type-II transmembrane protein Neurotactin (Nrt) and Wingless/Wnt (Nrt-Wg) is used to characterize the gradient-independent role of Wingless/Wnt (*Alexandre et al., 2014*; *Baena-Lopez et al., 2009*). Importantly, the mutant fly in which endogenous *Wg* was replaced with *Nrt-Wg* was viable with no major morphological defects (*Alexandre et al., 2014*). They also showed that the Nrt-Wg largely abrogated the gradient distribution of Wingless. These observations suggest that the gradient distribution of Wnt is not essential for normal development. We generated the mutant animal with membrane-tethered *lin-44* by inserting the codon-optimized Neurotactin (Nrt) and BFP into the endogenous *lin-44* locus by CRISPR/Cas9 (*miz56[nrt-bfp-lin-44]*) (*Figure 5A*). The BFP signal was observed on the membrane of the tail hypodermal cells including hyp10, suggesting that NRT-BFP-LIN-44 fusion proteins are localized on the surface of the *lin-44*-expressing cells (*Figure 5B*). The brighter BFP signal was observed at the interface between the hypodermal cells, likely because the signal comes from both hypodermal cells. Interestingly, we observed significant proportions of the *nrt-bfp-lin-44* animals have normal PDB structures with no posterior neurites at the L4 stage (*Figure 2D* and data not shown). Specifically, unlike *lin-44* null mutants, we observed normal pruning of the posterior neurites in *nrt-bfp-lin-44* animals (71.2%, 37/52) comparable to wildtype (*Figure 5C–5E*). This indicates that the membrane-tethered LIN-44 is sufficient to induce normal neurite pruning in PDB.

In addition to the traditional model of Wnt gradient formation by simple secretion and chemical diffusion, recent works have started to reveal the alternative modes of Wnt transport via cytonemes and extracellular vesicles (EVs) (*Gross et al., 2012*; *Routledge and Scholpp, 2019*; *Saha et al., 2016*; *Stanganello et al., 2015*). It is therefore possible that the membrane-tethered Wnt is still capable of making a gradient and reach the target cells from distant via these structures. To test if NRT-BFP-LIN-44 abolished the gradient distribution of LIN-44, we examined the phenotypes of two motor neurons (DA9 and DD6) that are known to be regulated by the LIN-44/Wnt gradient signal. The DA9 neuron is a cholinergic motor neuron whose cell body resides in the preanal ganglion. DA9 axon initially extends posteriorly along the ventral nerve cord and through commissure join the dorsal nerve cord where it then extends anteriorly to form *en passant* synapses onto dorsal body wall muscles (*Figure 6G*). Previous works have shown that LIN-44/Wnt gradient from the tail hypodermal cells inhibits synapse formation in the posterior axonal region of DA9, creating the synapse-free (or asynaptic) axonal domain (*Figure 6A and D*) (*Klassen and Shen, 2007*). In the *lin-44* null mutants, the ectopic synapse formation occurs in the dorsal posterior axonal domain of DA9 (*Figure 6B and D*). LIN-17::mCherry is localized within the asynaptic posterior axonal domain in a *lin-44*-dependent manner (*Figure 6A and B*, middle panels). Overexpression of *lin-44* from the *egl-20* promoter creates a larger asynaptic domain by displacing DA9 synapses more anteriorly, suggesting the LIN-44

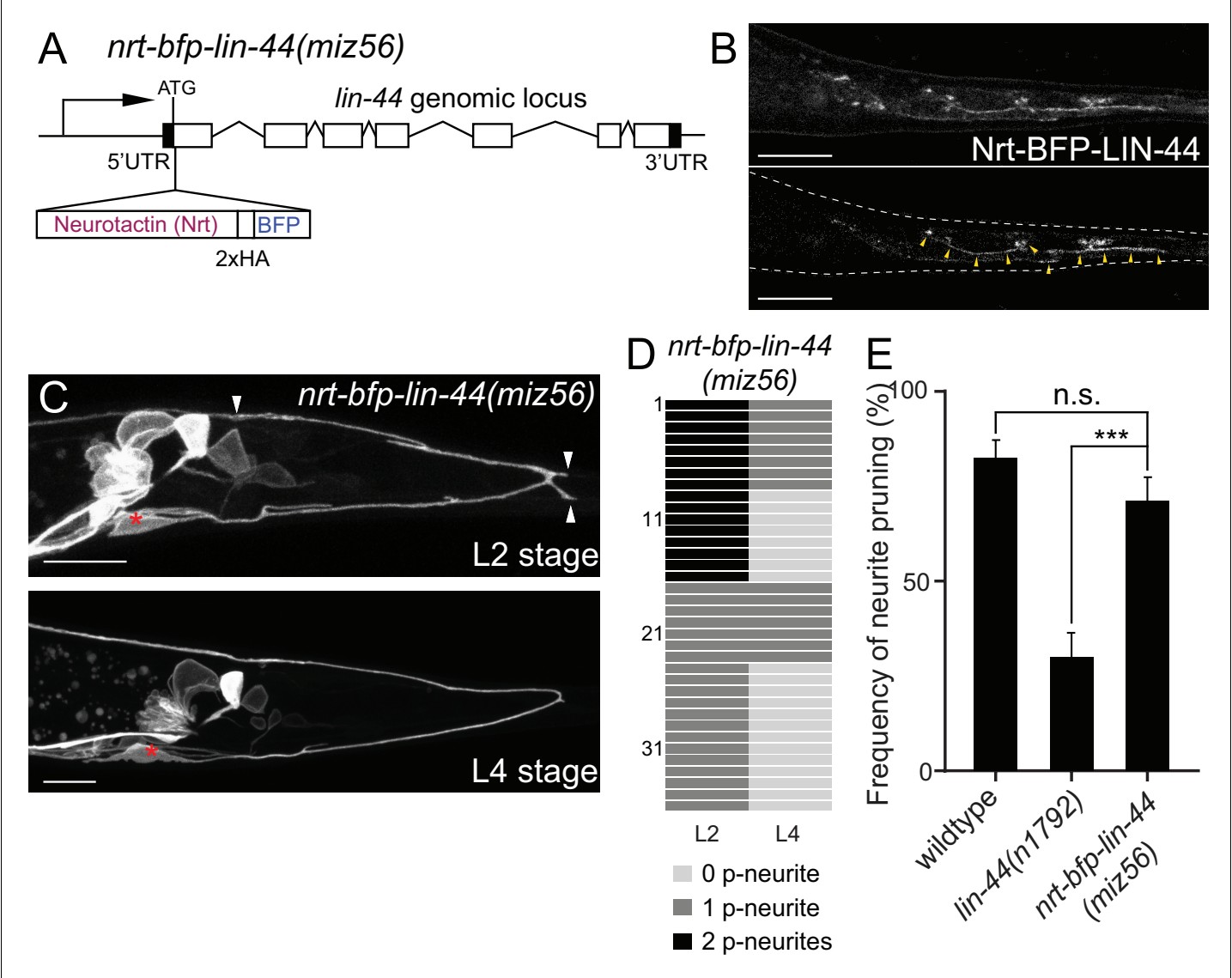

**Figure 5.** Membrane-tethered LIN-44 is sufficient to induce posterior neurite pruning in PDB. (**A**) A genomic structure of *lin-44* locus in *nrt-bfp-lin-44 (miz56)* mutants. (**B**) Subcellular localization of Nrt-BFP-LIN-44 in the adult animal. Maximum projection (top panel) and single plane (bottom panel) from z-stack images. Arrowheads denote Nrt-BFP-LIN-44 signal on the membrane. (**C**) Representative images of PDB structure labeled with *mizIs9* showing the pruning of posterior neurites in *nrt-bfp-lin-44(miz56)*. (**D**) Quantification of the posterior neurite numbers of 36 individual animals at L2 and L4 stages in *nrt-bfp-lin-44(miz56)* mutants. (**E**) Quantification of posterior neurite pruning frequency. ***p<0.001; n.s., not significant (Chi-square with Yates' correction). Error bars represent standard error of proportion (SEP). Scale bars: 10 μm.

gradient determines the length of the asynaptic domain (*Klassen and Shen, 2007*). In addition to the role of LIN-44 gradient in inhibiting synapse formation, the termination of the axonal outgrowth of the DD6 GABAergic motor neuron in the dorsal nerve cord is dependent on the LIN-44 gradient (*Maro et al., 2009*). In wildtype, the DD6 axon terminates before it reaches the rectum region (*Figure 6E and F*). In *lin-44* null mutants, it overextends beyond the rectum (*Figure 6E and F*). Over-expression of *lin-44* from the *lin-44* or *egl-20* promoters caused premature termination of the DD6 axon outgrowth (*Maro et al., 2009*). These two LIN-44 gradient-dependent phenomena were severely disrupted in *nrt-bfp-lin-44* animals and were indistinguishable from those in *lin-44* null mutants (*Figure 6C–6F*). These results strongly suggest that membrane-tethered LIN-44 is not capable of exerting the gradient-dependent function of LIN-44.

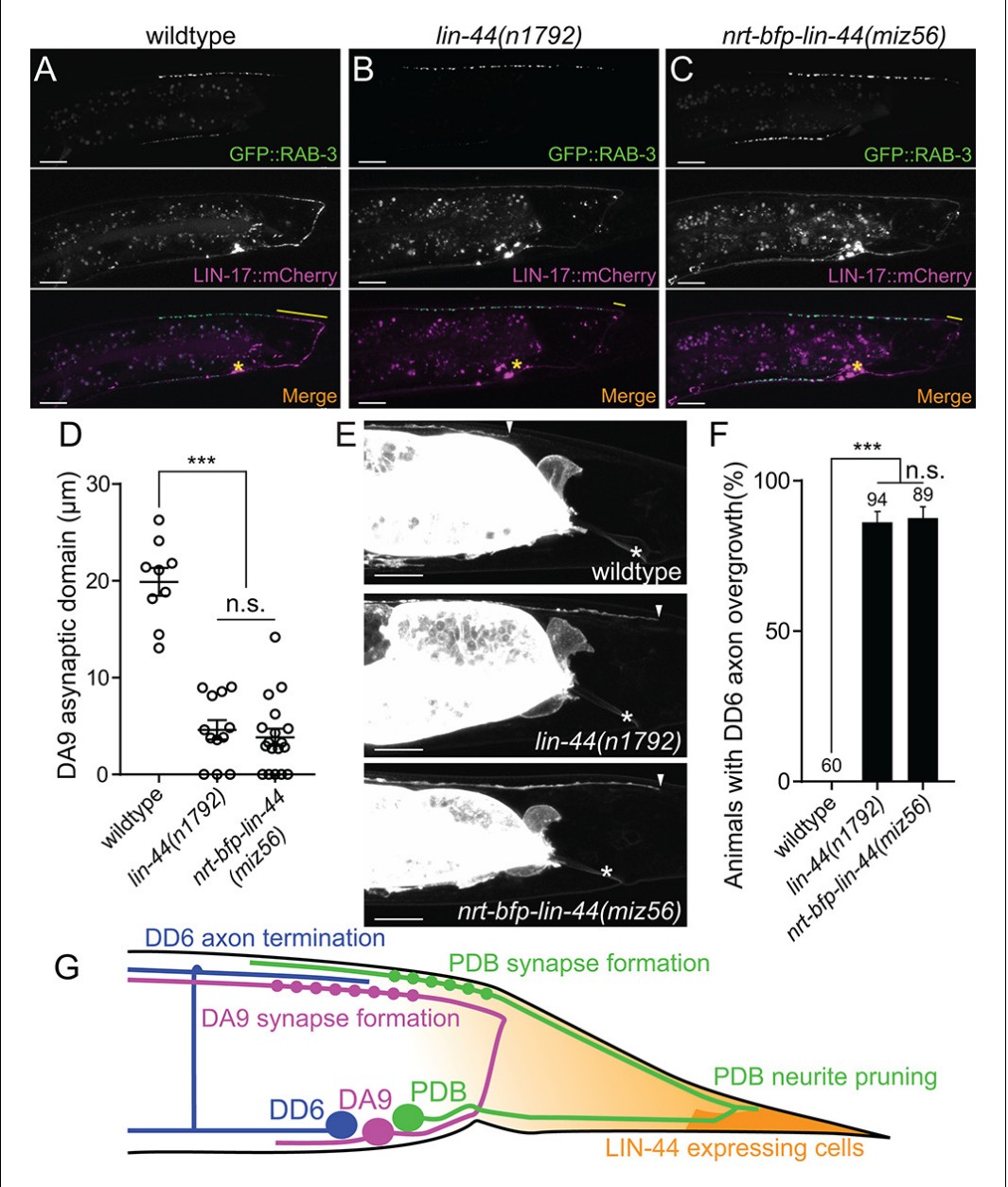

**Figure 6.** Membrane-tethered LIN-44 does not function as a gradient signal. (**A–C**) Representative images of DA9 presynaptic specializations labeled with GFP::RAB-3 (top panels), LIN-17::mCherry localization (middle panels) and merged images (bottom panels) in N2 (**A**), *lin-44(n1792)* (**B**) and *nrt-bfp-lin-44(miz56)* (**C**) animals. Asterisks denote DA9 cell body, and yellow lines represent the posterior asynaptic domain of the DA9 dorsal axon. GFP::RAB-3 puncta in the ventral side of the worm is due to the expression of P*mig-13::gfp::rab-3* in the VA12 motor neuron. (**D**) Quantification of the DA9 asynaptic domain length. The DA9 asynaptic domain is defined by the distance between the most posterior mCherry::RAB-3 puncta and the DA9 commissure. Each dot represents an individual animal. Error bars indicate mean ± SEM. \*\*\*p<0.001; n.s., not significant (one-way ANOVA). (**E**) Representative image of DD6 posterior axon in wildtype (top panel), *lin-44(n1792)* (middle panel) and *nrt-bfp-lin-44(miz56)* (bottom panel). Asterisks denote the position of the rectum; arrowheads denote the end of DD6 axon. (**F**) Quantification of DD6 axon overgrowth defect. Animals were considered as defective if DD6 axon terminal was located posteriorly to the rectum. \*\*\*p<0.001; n.s., not significant (Chi-square test). Error bars represent standard error of proportion (SEP). (**G**) Schematic of the relative positions of DD6, DA9, PDB neurons and *lin-44* expressing cells. Light graded orange represents hypothetical LIN-44 gradient. Scale bars: 10 μm.

The online version of this article includes the following source data and figure supplement(s) for figure 6:

*Figure 6 continued on next page*

*Figure 6 continued*

**Source data 1.** Quantification of DA9 and DD6 defects.
**Figure supplement 1.** LIN-44 gradient-dependent localization of RAB-3 in PDB.
**Figure supplement 1—source data 1.** Quantification of the RAB-3 puncta localization.

We further tested if NRT-BFP-LIN-44 is sufficient for replacing the function of diffusible LIN-44 in other aspects of PDB development. In *lin-44* null mutants, PDB exhibits defects in cell fate specification and neurite guidance (*Figure 2—figure supplement 1*). These defects are likely due to the lack of LIN-44 gradient signal since the PDB cell body is located away from the *lin-44*-expressing cells. The penetrance of these defects in PDB of *nrt-bfp-lin-44* was comparable to those of *lin-44* null mutants (*Figure 2—figure supplement 1*), suggesting that PDB requires both gradient-dependent and independent LIN-44/Wnt signal for normal development. Consistent with this idea, we found that the position of presynaptic RAB-3 puncta in the PDB neuron is dependent on the LIN-44 gradient. Similar to DA9, the PDB neuron has a large asynaptic domain in the dorsal neurite (*Figure 6—figure supplement 1*). In both *lin-44* null and *nrt-bfp-lin-44* animals, we observed ectopic mCherry::RAB-3 puncta in this neurite domain. This result suggests that PDB can respond to diffusible and non-diffusible Wnt signal in a context-dependent manner: it utilizes LIN-44 gradient signal to determine its cell fate, neurite guidance and the subcellular localization of the presynaptic vesicles, while its neurite pruning is induced by gradient-independent LIN-44 signal. *lin-44/wnt* instructs asymmetric localization of LIN-17/Fz and DSH-1/Dsh to the pruning neurites.

Wnt ligands activate intracellular signaling cascade by inducing the clustering of the receptors. Consistently, Frizzled receptors are localized to the site of action in Wnt-dependent manners. For example, LIN-17/Fz is localized asymmetrically on the cell cortex during asymmetric cell division, to the posterior neurite of the PLM mechanosensory neuron during neuronal polarization, and to the posterior asynaptic axonal domain to inhibit presynaptic assembly (*Goldstein et al., 2006*; *Hilliard and Bargmann, 2006*; *Klassen and Shen, 2007*). We therefore examined the subcellular localization of the LIN-17::GFP fusion protein during PDB posterior neurite pruning. We observed significantly higher accumulation of LIN-17::GFP puncta at the tip of the posterior neurites compared with those at the anterior neurite at L2 stage (*Figure 7A and D*). The asymmetric LIN-17::GFP localization at the posterior neurites is dependent on *lin-44/wnt*: we did not observe significant LIN-17::GFP puncta in the *lin-44* null mutant animals (*Figure 7B and D*). In contrast, LIN-17::GFP puncta were accumulated at the posterior neurites of PDB in the *nrt-bfp-lin-44* animals (*Figure 7C and D*). This is consistent with no obvious pruning defects in the *nrt-bfp-lin-44* animals. We note that the LIN-17::GFP puncta observed in the ventral neurite of wildtype animals were the absent in the *nrt-bfp-lin-44* animals (*Figure 7A and C*). These LIN-17::GFP puncta might be associated with the gradient-dependent function of *lin-44* such as neuronal polarization and guidance.

Dishevelled (Dsh/Dvl) proteins are the intracellular signaling components of both canonical and non-canonical Wnt signaling and are known to co-localize with active Frizzled receptors (*Wallingford and Habas, 2005*). There are three *Dishevelled* genes in *C. elegans* (*dsh-1*, *dsh-2* and *mig-5*). Similar to Frizzled receptors, Dsh proteins are often localized asymmetrically within the cells in Wnt-dependent manners (*Heppert et al., 2018*; *Klassen and Shen, 2007*; *Mizumoto and Sawa, 2007*; *Walston et al., 2004*). We did not observe significant pruning defects in *dsh-1* and *mig-5* single mutant animals, likely due to their functional redundancy (data not shown). Nevertheless, DSH-1/Dsh::GFP puncta were localized in the posterior neurites in the wildtype and *nrt-bfp-lin-44* but not in the *lin-44* null animals (*Figure 7—figure supplement 1*). Consistent with the idea that Dsh proteins are recruited to the active Frizzled receptors, DSH-1 localization at the posterior neurites was dependent on the LIN-17/Fz receptor (*Figure 7—figure supplement 1*). Taken these results together, we concluded that the locally restricted Wnt instructs asymmetric neurite pruning via recruiting Frizzled receptor and Dishevelled proteins to the pruning neurites (*Figure 8*).

## Discussion

Developmental neurite pruning plays crucial roles in shaping functional neurocircuits, yet not much is known about the signaling cues that instruct stereotyped neurite pruning during neuronal

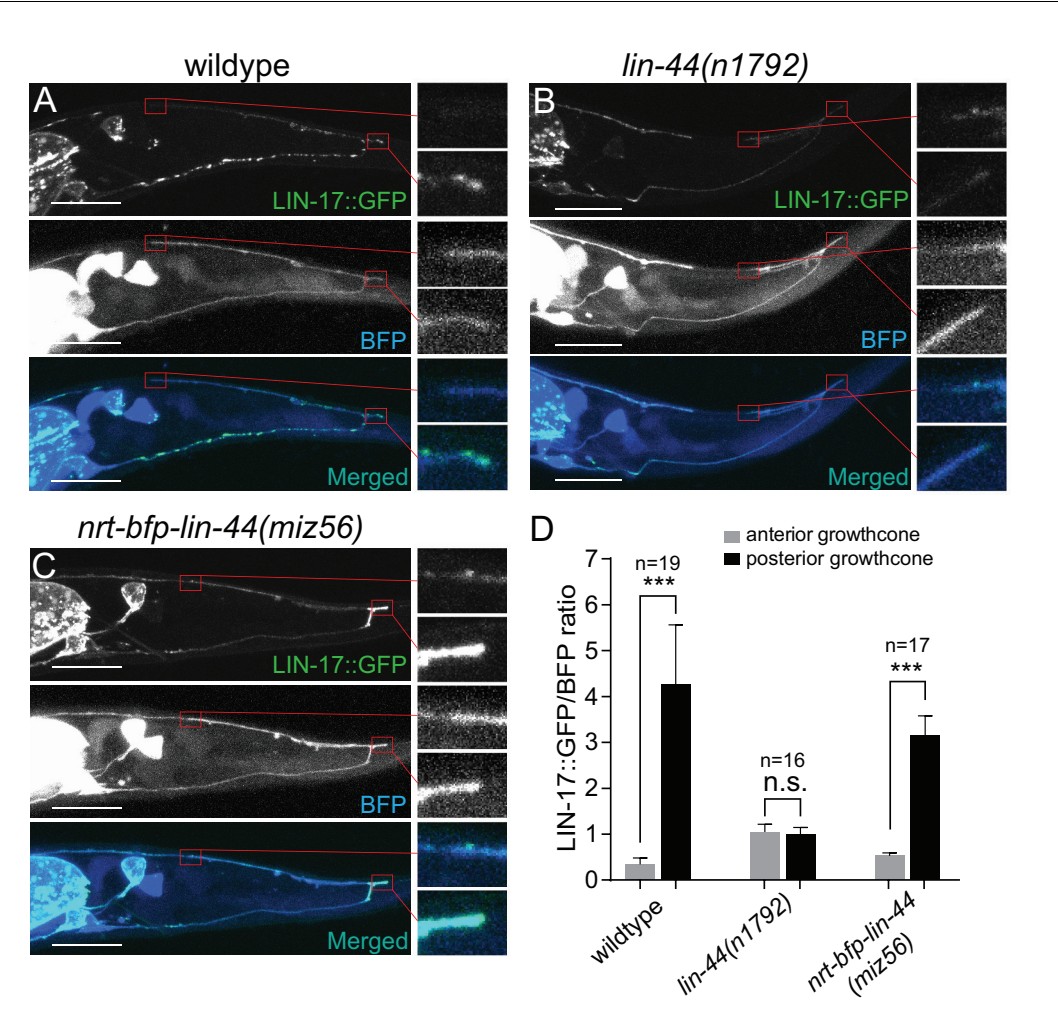

**Figure 7.** LIN-44/Wnt-dependent localization of LIN-17/Fz at the PDB posterior neurites. (**A–C**) Representative images of LIN-17::GFP localization (top panels), PDB neurite structure labeled with cytoplasmic BFP (middle panels) and merged images (bottom panels) in wildtype (**A**), *lin-44(n1792)* (**B**), and *nrt-bfp-lin-44(miz56)* (**C**) animals, respectively. Magnified images of the tip of anterior and posterior neurites are shown in the right panels. (**D**) Quantification of the normalized GFP/BFP signal ratio at the anterior and posterior growth cones. Error bars indicate mean ± SEM. ***p<0.001; n.s., not significant (Ratio paired t-test).

The online version of this article includes the following source data and figure supplement(s) for figure 7:

**Source data 1.** Quantification of the normalized GFP/BFP signal ratio at the anterior and posterior growth cones.

**Figure supplement 1.** Wnt-dependent localization of DSH-1/Dsh at the PDB posterior neurites.

**Figure supplement 1—source data 1.** Quantification of the normalized GFP/BFP signal ratio at the anterior and posterior growth cones.

development. Here, we found that Wnt signal from the hypodermal cells instructs stereotyped neurite pruning in the PDB cholinergic motorneuron in *C. elegans*. Asymmetric Wnt signal from the posterior hypodermal cells specifically induces pruning of the posterior neurite to sculpt the V-shape projection pattern of the PDB neurite. The unique structure of PDB and its topographic arrangement relative to the signaling cues allowed us to uncover the unique function of gradient-independent Wnt signaling in neurite pruning.

## Wnt signaling and neurite pruning

There have only been a handful of studies elucidating the roles of Wnt signaling in neurite pruning despite its critical roles in neuronal development and function. In *C. elegans*, two Wnts, CWN-1 and

CWN-2, act through CAM-1/Ror (receptor tyrosine kinase) to protect neurites from the *mbr-1*-mediated pruning in the AIM interneurons: the pruning defect in AIMs of *mbr-1* mutants is suppressed by mutations in *cam-1*, while overexpression of *cam-1* in AIMs inhibits neurite pruning (*Hayashi et al., 2009*). These data indicate that, in contrast to LIN-44/Wnt which instructs neurite pruning in PDB, CWN-1/2 inhibit neurite pruning in AIMs. It is possible that distinct receptors expressed in AIM and PDB neurons underlie the opposite effect of Wnt signaling on neurite pruning between them. Indeed, several studies have shown that *cam-1/Ror* and *lin-17/fz* function antagonistically (*Goh et al., 2012*; *Kidd et al., 2015*). The interesting analogy to our study has been observed in *Drosophila* contralaterally-projecting serotonin-immunoreactive deuterocerebral interneurons (CSDns) (*Singh et al., 2010*). CSDns undergo extensive dendritic remodeling during metamorphosis. The loss of Wingless and Wnt5 impairs the pruning of the transient dendritic branches of CSDn. Interestingly, the pruning of the CSDn dendrites is an activity-dependent process and requires upstream neurons. Wnt signaling functions downstream of the neuronal activity to eliminate dendrites with no synaptic inputs from the upstream neurons. While it remains to be determined whether Wnt signaling plays an instructive or a permissive role in CSDn dendrite pruning, these observations along with our present study show conserved role of Wnt signaling as a pruning cue.

Wnt and Frizzled receptors control distinct downstream signaling cascades (Wnt-β-catenin; Wnt-PCP; Wnt-Ca$^{2+}$) in context-dependent manners. We have examined potential downstream signaling components such as *bar-1/β-catenin*, *fmi-1/flamingo*, *vang-1/Van Gogh* but none of them showed significant neurite pruning defects (*Figure 2D*). Further candidate and forward genetic screenings will provide mechanistic insights into the Wnt-dependent neurite pruning.

## Gradient-independent Wnt signaling

*Drosophila* mutants whose endogenous *wingless* is replaced with the membrane-tethered Wingless (*nrt-wg*) are viable and exhibit no obvious morphological abnormalities. This implies that gradient-independent or contact-dependent Wnt signaling is largely sufficient for the normal tissue patterning during development (*Alexandre et al., 2014*). The authors also noted the importance of the gradient signal of Wingless since the *nrt-wg* fly showed delayed development and reduced fitness. Consistently, recent works revealed the critical requirement of secretion and diffusion of Wingless for renal tube patterning and intestinal compartmentalization in *Drosophila* (*Beaven and Denholm, 2018*; *Tian et al., 2019*).

In this study, we showed that the membrane-tethered LIN-44 (Nrt-BFP-LIN-44) is sufficient for the PDB neurite pruning but not for the synapse patterning. Nrt-BFP-LIN-44 is non-diffusible since it did not replace the function of LIN-44 in DD6 axon termination and DA9 synapse patterning. How can cells distinguish gradient-dependent and independent Wnt signaling? One possible explanation is that each Wnt-dependent process requires a distinct threshold of the Wnt concentration to activate

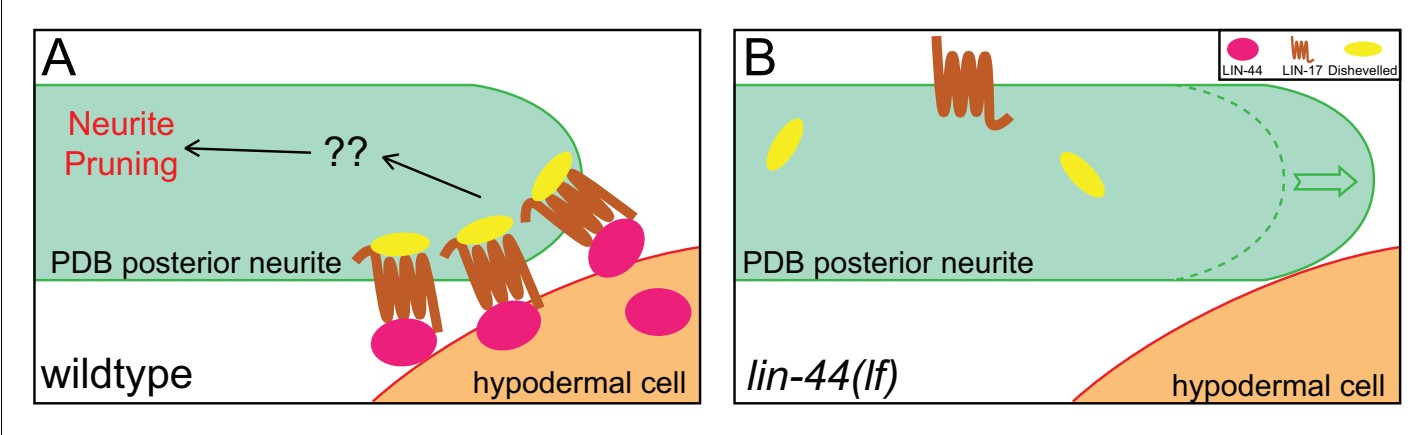

**Figure 8.** A model of neurite pruning in PDB. Local and high concentration of LIN-44/Wnt from the tail hypodermal cell (hyp10) induces PDB neurite pruning via recruiting LIN-17/Fz and DSH-1/Dsh in wildtype (left panel) and defective pruning in *lin-44* mutants (right panel).

the signaling cascade. The processes that are gradient-dependent might require lower Wnt concentration to execute the process compared with the gradient-independent process. Alternatively, it is also possible that the gradient-independent or contact-dependent processes require transmembrane co-factors that mediate contact-dependent Wnt signaling. *sdn-1/syndecan* was one such candidate as it has been shown that *sdn-1* is required for the Wnt-dependent spindle orientation in early embryonic development (*Dejima et al., 2014*), and *sdn-1* mutants exhibit PDB structure defects similar to *lin-44/wnt* (*Saied-Santiago et al., 2017*). However, we did not observe significant pruning defects in PDB of *sdn-1* mutants (data not shown). Further candidate and forward genetic screenings will reveal novel factors that are specifically required for the contact-dependent Wnt signaling.

The membrane-tethered LIN-44/Wnt (Nrt-LIN-44) efficiently induced posterior neurite pruning, suggesting the contact with Wnt-expressing cells rather than the gradient distribution of LIN-44 triggers the PDB neurite pruning. The contact-dependent neurite pruning model can explain why the anterior neurite, which is also located in close proximity to the *lin-44*-expressing cells, can ignore the LIN-44/Wnt pruning cue. It also explains how the PDB neurite can initially grow posteriorly from its cell body toward the source of pruning cue (LIN-44-expressing cells). On the other hand, we do not exclude the possibility that the anterior neurite responds to the attractive guidance cues from the anterior cells so that it is protected from the Wnt-dependent neurite pruning. We have tested several candidates including *sax-7/Neurofascin*, which guides and stabilizes the dendritic arbor structure of the PVD sensory neuron (*Dong et al., 2013*), *unc-129/TGFβ* which is a neurite attractant expressed from the postsynaptic dorsal body wall muscles (*Colavita et al., 1998*), *unc-6/Netrin* that acts as an axon attractant through *unc-40/DCC* receptor (*Chan et al., 1996*), and *cam-1/Ror* which acts downstream of Wnt to protect neurites from pruning (*Hayashi et al., 2009*), but none of them showed ectopic pruning of the anterior neurites (data not shown). Future studies will be necessary to determine the threshold of Wnt concentration to induce neurite pruning as well as potential mechanisms that protect anterior neurite from pruning.

## Materials and methods

**Key resources table**

| Reagent type (species) or resource | Designation | Source or reference | Identifiers | Additional information |
|---|---|---|---|---|
| Gene (*C. elegans*) | *lin-44* | NA | E01A2.3 | |
| Gene (*C. elegans*) | *lin-17* | NA | Y71F9B.5 | |
| Strain, strain background (*C. elegans*) | *lin-44(n1792)* | *C. elegans* stock center (CGC) | MT5383 | |
| Strain, strain background (*C. elegans*) | *lin-17(n671)* | *C. elegans* stock center (CGC) | MT1306 | |
| Strain, strain background (*C. elegans*) | *lin-44(miz56)* | This study | UJ1124 | *nrt-bfp-lin-44* |
| Strain, strain background (*C. elegans*) | *mizIs9* | This study | UJ261 | PDB neurite/ synaptic marker |

## Strains

Bristol N2 strain was used as wildtype reference. All strains were cultured in the nematode growth medium (NGM) as described previously (*Brenner, 1974*) at 22°C. The following alleles were used in this study: *lin-44(n1792)*, *lin-44(miz56)*, *lin-17(n671)*, *egl-20(n585)*, *mig-1(e1787)*, *cam-1(gm122)*, *bar-1 (ga80)*, *fmi-1(tm306)*, *vang-1(ok1142)*, *unc-13(e450)*. Genotyping primers are listed in the supplemental material.

## Transgenes

The transgenic lines were generated using standard microinjection method (*Fire, 1986*; *Mello et al., 1991*): *mizIs9* (P*kal-1::zf1-GFPnovo2::CAAX*; P*vha-6::zif-1*; P*kal-1::mCherry::rab-3*); *wyIs486* (P*flp-13::2xGFP*, P*plx-2::2xmCherry*; P*odr-1::RFP*); *mizEx248*, *mizEx389* (P*lin-44::lin-44*; P*odr-1::RFP*); *mizEx349*, *mizEx350* (P*lin-44::egl-20*; P*odr-1::RFP*); *mizEx384*, *mizEx385* (P*egl-20::lin-44*, P*odr-1::RFP*); *mizEx380*, *mizEx381* (P*unc-129dm::lin-44*, P*odr-1::RFP*); *mizEx386*, *mizEx387* (P*kal-1::lin-17*; P*odr-1::RFP*), *mizEx374*, *mizEx376* (P*rgef-1::lin-17*; P*odr-1::RFP*); *mizEx366* (P*mig-13::GFP::rab-3*; P*mig-13::lin-17::mCherry*; P*odr-1::GFP*); *mizEx271* (P*lin-44::BFP*; P*odr-1::GFP*); *mizEx295* (P*kal-1::lin-17::GFPnovo2*; P*kal-1::bfp*; P*odr-1::GFP*); *mizEx291* (P*kal-1::dsh-1::GFPnovo2*; P*kal-1::bfp*; P*odr-1::GFP*). All rescuing constructs were injected at 10 ng/µl except P*unc-129dm::lin-44* which was injected at 20 ng/µl.

## Plasmid construction

*C. elegans* expression clones were made in a derivative of pPD49.26 (A. Fire), the pSM vector (a kind gift from S. McCarroll and C. I. Bargmann). *dsh-1* cDNA clone was obtained by RT-PCR from N2 mRNA using Superscript III First-strand synthesis system and Phusion High-Fidelity DNA Polymerase (Thermo Fisher Scientific). *lin-17* cDNA clone was obtained from the plasmid used in the previous work (*Mizumoto and Shen, 2013*).

### P*kal-1::zf1-GFPnovo2-caax* plasmid

The 3.7 kb fragment of *kal-1* promoter was amplified from N2 genomic DNA using Phusion high fidelity enzyme (ThermoFisher Scientific, USA) and cloned into the *Sph*I and *Asc*I sites of the pSM-GFPnovo2 vector (*Hendi and Mizumoto, 2018*). 111 bp sequence of the ZF1 zinc finger domain from *pie-1* and 51 bp sequence of the CAAX sequence from human KRas were inserted into the 5' and 3' of GFPnovo2 sequence, respectively, using Gibson assembly method (*Armenti et al., 2014*; *Gibson et al., 2009*). The intestinal GFP signal was reduced by expressing *zif-1* in the intestine under the *pha-6* promoter to degrade ZF1-GFPnovo2::CAAX.

### *Neurotactin-BFP-lin-44* repair plasmid

The plasmid containing a codon-optimized *neurotactin* cDNA with 2xHA tag was obtained from GeneArt (ThermoFisher Scientific, USA). *C. elegans* codon-optimized *BFP* with three synthetic introns was inserted into the 3' end of the Neurotactin-2xHA sequence to generate *Neurotactin-2xHA-BFP* construct. 700 bp of the 5' homology arm of the *lin-44* promoter region, *Neurotactin-2xHA-BFP* (5' portion until middle of intron 1), *BFP* (3' portion from intron 1), 700 bp of the 3' homology arm spanning *lin-44* coding region are cloned into the *Sac*II and *Not*I sites of a dual-marker selection cassette (*loxP* + P*myo-2::GFP::unc-54* 3'UTR + P*rps-27::neoR::unc-54* 3'UTR + *loxP* vector) (*Au et al., 2019*; *Gibson et al., 2009*; *Norris et al., 2015*).

### *lin-44* gRNA constructs

Two 19 bp gRNA sequences (gRNA3-CGATCAGTGGTGCACCTGC, gRNA4-TATTTCCGTCTTCAGC-CAA) near the start codon of the *lin-44* gene were selected using CRISPR guide RNA selection tool (http://genome.sfu.ca/crispr/) and were cloned into *Rsa*I site of the sgRNA (F+E) vector, pTK73 (*Obinata et al., 2018*).

## CRISPR

The repair template plasmid, two *lin-44* sgRNA plasmids and a Cas9 plasmid (Addgene# 46168) (*Friedland et al., 2013*) were co-injected into young adults. The candidate genome-edited animals were screened based on G418 resistance and uniform expression of P*myo-2::GFP* in the pharynx as described previously (*Au et al., 2019*). The selection cassette was excised by injecting Cre recombinase plasmid (pDD104, Addgene #47551). Excision of the selection cassette, which was inserted within the first intron of BFP, reconstituted *Neurotactin-2xHA-BFP-lin-44*. The junctions between *neurotactin* and *BFP* as well as *BFP* and *lin-44* coding sequence were confirmed by Sanger sequencing.

## Semi-synchronization of *C. elegans*

Eggs at various stages were collected by bleaching the gravid adults with a bleaching solution (2.5% sodium hypochlorite and 0.5N Sodium Chloride) for 5 min. The eggs were washed twice with M9 buffer and seeded onto an NGM plate. For observing the pruning event, the animal was rescued from the slide after the first imaging at the L2 stage (approximately 26 hr after seeding eggs onto the NGM plates) and placed it in an individual NGM plate to resume the development for 24 hr before re-imaging at early L4 stages.

## Confocal microscopy

Images of fluorescently tagged fusion proteins were captured in live *C. elegans* using a Zeiss LSM800 Airyscan confocal microscope (Carl Zeiss, Germany). Worms were immobilized on 2% agarose pad using a mixture of 7.5 mM levamisole (Sigma-Aldrich) and 0.225M BDM (2,3-butanedione monoxime) (Sigma-Aldrich). Images were analyzed with Zen software (Carl Zeiss) or Image J (NIH, USA).

For time-lapse imaging, animals were mounted onto 5% agarose pads and immobilized using 0.5 µl of 0.10 µmpolystyrene latex microsphere (Alfa Aesar # 427124Y). The coverslip was sealed with Vaseline (Vaseline Jelly Original), to avoid dehydration of the animals and agarose pad during imaging.

For quantification of LIN-17::GFP and DSH-1::GFP signal at the growth cones, 1.34 µm × 0.37 µm (LIN-17::GFP) or 1.54 µm × 0.46 µm (DSH-1::GFP) region of interest (ROI) were set at the tip of anterior and posterior neurites. The signals of the GFP and BFP channels from the adjacent region with the same size of ROI was used to subtract the background signal. The GFP signal was normalized by the BFP signal of the same ROI.

## Statistics

Data were processed using Prism7 (GraphPad Software, USA). We applied the Chi-square test (with Yates' continuity corrected) for comparison between two binary data groups, and one-way ANOVA method for comparison among more than three parallel groups with multiple plotting points. Data were plotted with error bars representing standard errors of the mean (SEM) or standard error of the proportion (SEP). *, ** and *** represent p-value<0.05,<0.01 and<0.001 respectively.

## Acknowledgements

We are grateful to Don Moerman for suggestions and strains. We thank Harald Hutter, Kenji Sugioka and Riley St. Clair for critical reading of the manuscript, Ardalan Hendi for generating some strains, Shinsuke Niwa for the pTK73 plasmid, and the Mizumoto lab members for general discussions. We also thank Zhenhao Guo and members in the Chalfie lab for comments and suggestions on the manuscript deposited on bioRxiv. Some strains used in this study are obtained from the CGC, which is funded by NIH Office of Research Infrastructure Programs (P40 OD010440), and from the National Bioresource Project, Japan. This project is funded by HFSP (CDA-00004/2014) and NSERC (RGPIN-2015–04022). KM is a Tier 2 Canada Research Chair and a Michael Smith Foundation for Health Research scholar.

## Additional information

### Funding

| Funder | Grant reference number | Author |
| --- | --- | --- |
| Natural Sciences and Engineering Research Council of Canada | RGPIN-2015-04022 | Kota Mizumoto |
| Human Frontier Science Program | CDA-00004/2014 | Kota Mizumoto |

The funders had no role in study design, data collection and interpretation, or the decision to submit the work for publication.

## Author contributions
Menghao Lu, Formal analysis, Validation, Investigation, Visualization, Methodology, Writing—review and editing; Kota Mizumoto, Conceptualization, Formal analysis, Supervision, Funding acquisition, Validation, Investigation, Visualization, Methodology, Writing—original draft, Writing—review and editing

## Author ORCIDs
Menghao Lu ⓘD http://orcid.org/0000-0001-7770-6317
Kota Mizumoto ⓘD https://orcid.org/0000-0001-8091-4483

## Decision letter and Author response
Decision letter https://doi.org/10.7554/eLife.50583.sa1
Author response https://doi.org/10.7554/eLife.50583.sa2

# Additional files

## Supplementary files
• Transparent reporting form

## Data availability
All data generated or analyzed during this study are included in the manuscript.

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

## Appendix 1

# Supplementary information

**Genotyping primers**

| | |
|---|---|
| **cam-1(gm122):** | |
| Forward: | aacttacactctgaagcgccgtgc |
| Reverse: | ccaacttcagatgatggatcg |
| **egl-20(n585):** | wildtype PCR product can be digested with *HpyCH4V:* |
| Forward: | cttacctctcaaatttgaacttattcttgc |
| Reverse: | cctcattaccattcaactgatag |
| **fmi-1(tm306):** | |
| Forward: | gtgaaattacatgttaactgagg |
| Reverse: | tcgcctacaagaagtaacttacatg |
| **lin-44(n1792):** | *wildtype PCR product can be digested with NcoI:* |
| Forward: | gtgcgaatcgtttgagatttcagccatg |
| Reverse: | catctggttgttacacgcacaatcg |
| **mig-1(e1787):** | mutant PCR product can be digested with *BglII:* |
| Forward: | acagcacaaaattcaaagccctc |
| Reverse: | atttttgagccattcaaaaagaattttgaagatc |
| **sax-7(nj48):** | |
| Forward: | tgagatgaaagaaggaggagtgc |
| Reverse: | cacacacaatggcgcacaag |
| **lin-17(n671):** | wildtype PCR product can be digested with *MfeI:* |
| Forward: | ccgcatttttcgtagatcacaccg |
| Reverse: | actgttgtttacagtcaattgtcattcgggtcaatt |
| **vang-1(ok1142):** | |
| Forward: | accttgacaacgccagacag |
| Mutant reverse: | gcggcaattaggcgatacctg |
| Wildtype reverse: | cgagcgaagtttgagtgaatc |
| **dsh-1(ok1445):** | |
| Forward: | cttgagatagccctgcaagac |
| Reverse: | gctccaccacttgctaagattg |
| **mig-5(tm2639):** | |
| Forward: | aagcattcgctcctcatcatc |
| Reverse: | atctcgacgatgaaacgactc |
| **nrt-bfp-lin-44(miz56):** | |
| Wildtype forward: | ggaaatagtgtgtggatgag |
| Mutant forward: | aaggaaccgtggacaaccatc |
| Reverse: | ccactgatcgtcgggatctc |
| **sdn-1(ok449):** | |
| Forward: | agaagtgtctcgttcagtgg |
| Wildtype reverse: | ctgcaaagccgacaacggtac |

continued

**Genotyping primers**

| | |
|---|---|
| Mutant reverse: | gagatgccggtcaggtgattac |
| *cwn-1(ok546):* | |
| Forward: | agctggctaggtcccagaag |
| Reverse: | tcgaacgatcctcttcagtac |

