## [Decision Letter]

**Acceptance summary:**

Your study very nicely describes the molecular basis of an intriguing, developmental neurite pruning process, a topic of broad and general interest in the developmental neuroscience community.

**Decision letter after peer review:**

Thank you for submitting your article "Gradient-independent Wnt signaling instructs asymmetric neurite pruning in *C. elegans*" for consideration by *eLife*. Your article has been reviewed by two peer reviewers, and the evaluation has been overseen by a Reviewing Editor (Oliver Hobert) and Marianne Bronner as the Senior Editor. The reviewers have opted to remain anonymous. The reviewers have discussed the reviews with one another and with the Reviewing Editor.

There is general agreement about the overall interest of this story and the novelty of insights provided. However, everyone agreed that a small and well-defined number of very straight-forward and simple experiments need to be done that will solidify the conclusions of the manuscript. The experiments are detailed below in the reviewer's reports.

Reviewer #1:

The authors describe a role of Wnt signaling in the pruning of ectopic neurites of a developing neuron in *C. elegans*. Interestingly, they find that this is dependent on close contact between the outgrowing neuronal process and the Wnt producing cells, and using membrane-tethered Wnt, they show that short-range signaling is sufficient for pruning. Moreover, they show that this local Wnt activity induces clustering of Frizzled and Dishevelled at the ectopic neurite. The role of Wnt signaling in neurite pruning is still poorly understood and this study provides important new insight into this process.

I have one major concern that the authors should address, and that is the role of *lin-44/Wnt* in PDB cell fate specification. In *lin-44* mutants, 32.3% of animals show defects in PDB development, ranging from absence of the cell to neuronal polarity and axon guidance defects. It is therefore important to show that the effect on pruning is distinct from these functions in cell fate specification. One way to show this would be to examine these other aspects of PDB development in the membrane-tethered LIN-44 strain. If membrane-tethered LIN-44 only rescues the pruning phenotype, this would be a strong argument that this is indeed a separate (short-range) function of LIN-44.

Reviewer #2:

The manuscript by Lu and Mizumoto entitled "Gradient-independent Wnt signaling instructs asymmetric neurite pruning in *C. elegans*" characterizes the function of the Wnt ligands and Frizzled receptors in regulating neurite pruning. In particular, the authors first identify a novel pruning event on the axon of the posteriorly located PDB neuron, occurring during *C. elegans* larval stages. Then, they identify Wnt/LIN-44 as a key ligand for correct pruning of posterior extra branches of the PDB neurite. Moreover, the authors reveal a PDB-cell-autonomous function for LIN-17/Frizzled, which functions as the receptor for LIN-44 in this process. Interestingly, they also show that in this context LIN-44 can function when membrane-bound in the secreting epidermal cells, revealing a short-distance role in this process. Finally, they show an effect for LIN-44 in recruiting LIN-17 and Dishevelled to the posterior pruned sections of PDB. The significance of the manuscript relies on the identification of a novel pruning event on a specific neuron, and on the characterization of the molecular mechanisms regulating it. The manuscript is generally well written with a nice flow of the rationale of the experiments, and the discovery itself is quite compelling. However, the enthusiasm is reduced because a number of key experiments are underdeveloped to fully support the conclusions (either by a limited number of animals, a limited number of strains tested, or both). Here below is a detailed description of these considerations. When completed, these experiments will strengthen the conclusions of this manuscript.

1) Figure 1B, and D. The number of animals examined is too low. In general, for morphological studies, at least 50 animals are examined (see also those studies extensively cited by the authors, such as Klassen and Shen, 2006). The n should be added to the individual bars.

2) In Figure 1—figure supplement 1 (time lapse) there is an additional branching more anterior to the one from which the posterior pruned sections extend. This branch, visible in every section, gets resolved at the 80 min time-point. This is not discussed anywhere in the manuscript. Is it a normal occurrence, or only observed in this representative animal? If common, is it affected by the Wnt and Frizzled mutations?

3) Figure 2E. n are low and some of the genes tested here have not been discussed in the paper.

4) Figure 3D. n too low to determine any difference between *lin-44* and *lin-17*, as well as any significant effect of the double mutants. This is important, because a difference between *lin-17* and *lin-44* might indicate that another molecule, possibly another Wnt, is involved. As this regard, have the author tested *cwn-1*, which is also expressed in the posterior region of the animal at early stages of development?

5) In Figure 3—figure supplement 1B, n should be indicated in each bar. How many transgenic strains were analyzed here? What DNA concentrations? Is there a significant difference in the rescue obtained with the rgef-1 promoter versus that provided by *kal-1*? If so, is it possible that there is also additional, non-cell-autonomous function of LIN-17?

6) In Figure 4. The data showing that P*egl-20::lin-44* does not rescue the *lin-44* lack of pruning is weak: only one transgenic strain was tested with an n = 22. To support this conclusion, the authors should show the absence of rescue for at least three independent extrachromosomal array transgenic strains, and for different concentrations of the transgene. Similarly, the data supporting the functional divergence between LIN-44 and EGL-20 is weak. Results showing that *Plin-44::egl-20* does not rescue the *lin-44* lack of pruning are only for a very small sample size (n = 14) and for only one transgenic strain. This hypothesis should be further tested.

7) Figure 5. The data showing that there is "normal" pruning in *nrt-bfp-lin-44* animals should be strengthen by increasing the sample size to at least 50 animals. The difference in pruning rate between *nrt-bfp-lin-44* and wild-type animals (65.6% vs. 82.1%) could indicate that the membrane-tethered version cannot fully replace the secreted version.

8) The idea of an instructive, highly concentrated source of LIN-44 is valid. It will be important to test if other transgenic strains expressing LIN-44 under the Pegl-20 promoter do present pruning of the anterior neurites (see point 6 above). Did they try to use an *unc-129* promoter to express LIN-44 in a region in which the anterior/dorsal PDB neurite encounters the dorsal muscle?

---

## [Author Response]

Reviewer #1:

[…] I have one major concern that the authors should address, and that is the role of lin-44/Wnt in PDB cell fate specification. In lin-44 mutants, 32.3% of animals show defects in PDB development, ranging from absence of the cell to neuronal polarity and axon guidance defects. It is therefore important to show that the effect on pruning is distinct from these functions in cell fate specification. One way to show this would be to examine these other aspects of PDB development in the membrane-tethered LIN-44 strain. If membrane-tethered LIN-44 only rescues the pruning phenotype, this would be a strong argument that this is indeed a separate (short-range) function of LIN-44.

We appreciate the reviewer for raising this point. It is very important to test if the membrane-tethered LIN-44 exclusively rescues the pruning defects of the posterior neurites in PDB but not the other gradient-dependent phenotypes. *nrt-bfp-lin-44* exhibited cell fate determination and axon guidance defects at a similar frequency as *lin-44* null mutants (*lin-44:*30%, n=100; *nrt-bfp-lin-44*: 30.1%, n=103). The quantification is now included in Figure 2—figure supplement 1. In addition, we also found that the localization of the presynaptic marker, mCherry::RAB-3, in PDB is equally affected in *lin-44* null and *nrt-bfp-lin-44* animals. mCherry::RAB-3 puncta are mislocalized in the posterior dorsal neurite of the PDB neuron. This phenotype is very similar to the synapse localization defects observed in the DA9 neuron of *lin-44* null and *nrt-bfp-lin-44* animals. We have included this observation in Figure 6—figure supplement 1, and added the following sentences in the text.

“We further tested if NRT-BFP-LIN-44 is sufficient for replacing the function of diffusible LIN-44 in other aspects of PDB development. […] This result suggests that PDB can respond to diffusible and non-diffusible Wnt signal in a context-dependent manner: it utilizes LIN-44 gradient signal to determine its cell fate, neurite guidance and the subcellular localization of the presynaptic vesicles, while its neurite pruning is induced by gradient-independent LIN-44 signal.”

Reviewer #2:

[…] The manuscript is generally well written with a nice flow of the rationale of the experiments, and the discovery itself is quite compelling. However, the enthusiasm is reduced because a number of key experiments are underdeveloped to fully support the conclusions (either by a limited number of animals, a limited number of strains tested, or both). Here below is a detailed description of these considerations. When completed, these experiments will strengthen the conclusions of this manuscript.

We thank the reviewer for the criticism. We agree that the sample number for some of the experiments were low to make solid conclusions. We describe the details below.

1) Figure 1B, and D. The number of animals examined is too low. In general, for morphological studies, at least 50 animals are examined (see also those studies extensively cited by the authors, such as Klassen and Shen, 2006). The n should be added to the individual bars.

We appreciate your criticism. We did our best to increase the sample size. We would like to mention that unlike quantifications in other papers we cite, which mostly use a single snapshot of the animals, our quantification requires a lot more steps, which made the quantification low throughput. For examining the pruning event of single animal, we need to first look for an L2 animal with posterior neurites, take confocal image, rescue the animal from the slide, let it grow for a day and re-take confocal image 24 hours later. During this process, we lose more than half of the animals as they often do not recover or do not mount on the slide in proper orientation.

2) In Figure 1—figure supplement 1 (time lapse) there is an additional branching more anterior to the one from which the posterior pruned sections extend. This branch, visible in every section, gets resolved at the 80 min time-point. This is not discussed anywhere in the manuscript. Is it a normal occurrence, or only observed in this representative animal? If common, is it affected by the Wnt and Frizzled mutations?

Yes, there are neurites that emerge from the anterior neurites that also undergo pruning. We think their pruning is also *lin-44/wnt*-dependent since we often observe multiple neurites at L4 stage of the *lin-44* mutant but not in the wildtype animals. However, we did not conduct detailed analyses of the pruning of these neurites since their position and the timing of the pruning are not as stereotyped as the posterior neurites. We have added arrowheads to indicate these transient neurites in the figure and the description in the figure legend.

3) Figure 2E. n are low and some of the genes tested here have not been discussed in the paper.

We have examined at least 100 animals per genotype, and removed the genotypes we did not mention in the text. The total sample number including the animals with axon guidance and cell fate defects is included in Figure 2—figure supplement 1 B.

4) Figure 3D. n too low to determine any difference between lin-44 and lin-17, as well as any significant effect of the double mutants. This is important, because a difference between lin-17 and lin-44 might indicate that another molecule, possibly another Wnt, is involved. As this regard, have the author tested cwn-1, which is also expressed in the posterior region of the animal at early stages of development?

We have increased the sample number as far as we could, and still observed no significant differences among *lin-44; lin-17, lin-44* and *lin-17* mutants.

“The pruning defect in the *lin-44; lin-17* double mutants (14.3%: 6/42 of the posterior neurites were pruned) was not significantly different from *lin-17* single mutants, suggesting that *lin-44* and *lin-17* act in the same genetic pathway (Figures 3C and 3D).”

We also examined the PDB morphology in *cwn-1* mutants and we did not observe obvious pruning defects. We included this data in Figure 2D and added the following description in the text.

“Another Wnt gene, *cwn-1*, is also expressed in the posterior region of the worm (Harterink et al., 2011). The PDB morphology was indistinguishable from wildtype in the *cwn-1* mutants (Figure 2D and data not shown). While we do not completely exclude the involvement of other Wnts, our results indicate that LIN-44 is the major instructive cue for PDB neurite pruning.”

5) In Figure 3—figure supplement 1B, n should be indicated in each bar. How many transgenic strains were analyzed here? What DNA concentrations? Is there a significant difference in the rescue obtained with the rgef-1 promoter versus that provided by kal-1? If so, is it possible that there is also additional, non-cell-autonomous function of LIN-17?

We now included the second transgenic line for each rescue experiment, and confirmed that they showed similar to the same rescue activity compared with the first line. We also observed multiple independent transgenic lines for all transgenic lines at L4 stage to make sure the consistency of the experiments (data not shown).

Since neither promoter is PDB specific, we cannot completely exclude the possible non-cell autonomous roles of *lin-17* in PDB neurite pruning. The slightly weaker rescue activity of P*kal-1::lin-17* could be due to the late onset of the *kal-1* promoter in the PDB neuron. However, together with the specific and *lin-44-*dependent subcellular localization of LIN-17 at the pruning neurite of PDB, we believe that *lin-17* functions cell autonomously in PDB.

Regarding DNA concentration of the rescuing plasmids, we injected at 10ng/µl. This information is now included in the ‘Transgenes’ section in the Materials and methods as follows:

“All rescuing constructs were injected at 10ng/µl except P*unc-129dm::lin-44* which was injected at 20ng/µl.”

6) In Figure 4. The data showing that Pegl-20::lin-44 does not rescue the lin-44 lack of pruning is weak: only one transgenic strain was tested with an n = 22. To support this conclusion, the authors should show the absence of rescue for at least three independent extrachromosomal array transgenic strains, and for different concentrations of the transgene. Similarly, the data supporting the functional divergence between LIN-44 and EGL-20 is weak. Results showing that Plin-44::egl-20 does not rescue the lin-44 lack of pruning are only for a very small sample size (n = 14) and for only one transgenic strain. This hypothesis should be further tested.

We have included the second independent transgenic lines and confirmed that neither P*egl-20::lin-44* nor *Plin-44::egl-20* rescued the pruning defect of *lin-44* mutants. While we could not conduct three independent lines due to the time-consuming nature of the experiment, we obtained more than three independent lines for each construct, and *lin-44* animals carrying these transgenes looked like *lin-44* mutants at L4 stage (data not shown). We also conducted another experiment to express *lin-44* from the *unc-129dm* dorsal body wall muscles-specific promoter. Neither of two independent lines of *lin-44* mutants with *Punc-129dm::lin-44* rescued the pruning defects of the PDB posterior neurites. The data is included in Figure 4 and we have added the following sentence in the text:

“Similarly, ectopic anterior expression of *lin-44* from the dorsal body wall muscles using the truncated *unc-129* promoter (*Punc-129dm*) (Colavita et al., 1998) was not sufficient to rescue the PDB pruning defect in *lin-44* null mutants (Figure 4A and 4B).”

With these data, we conclude that the anterior expression of *lin-44* is not sufficient to rescue the posterior neurite pruning of PDB.

We could not inject these Wnt constructs at high concentrations since they cause severe morphological defects which indirectly affect the PDB neurite structure.

7) Figure 5. The data showing that there is "normal" pruning in nrt-bfp-lin-44 animals should be strengthen by increasing the sample size to at least 50 animals. The difference in pruning rate between nrt-bfp-lin-44 and wild-type animals (65.6% vs. 82.1%) could indicate that the membrane-tethered version cannot fully replace the secreted version.

We have increased the sample size and saw no difference between wildtype and *nrt-bfp-lin-44* mutants (82.6% and 71.2%). The slight, non-significant reduction in pruning rate in the *nrt-bfp-lin-44* animal could be due to the slight axon guidance defects we could not detect.

8) The idea of an instructive, highly concentrated source of LIN-44 is valid. It will be important to test if other transgenic strains expressing LIN-44 under the Pegl-20 promoter do present pruning of the anterior neurites (see point 6 above). Did they try to use an unc-129 promoter to express LIN-44 in a region in which the anterior/dorsal PDB neurite encounters the dorsal muscle?

We have expressed *lin-44* from the *unc-129dm* promoter in the wildtype background, and observed no ectopic pruning of the anterior neurite (line #1, n=96; line #2, n=45). Interestingly, the PDB morphology was indistinguishable from wildtype: we did not even observe axon guidance defects in the PDB neuron in these animals. The normal PDB morphology or lack of ectopic pruning of the anterior neurite could be due to the ectopic expression of LIN-44 in the body wall muscles. As the body wall muscles don’t normally express LIN-44, they might not secrete enough amount LIN-44 to induce neurite pruning. It is also important to note that muscle cells do not physically in contact with the neurites in the dorsal nerve cord until muscle arms (gigantic postsynaptic spine-like protrusions) reach the neurites to form neuromuscular junctions. The growth cone might not contact with these muscle cells to induce pruning. Since there can be many reasons for why such an artificial condition did not cause expected gain-of-function phenotype (ectopic anterior neurite pruning), we decided not to include this data in the manuscript.